# Chemical Composition and Cytoprotective Activities of Methanolic Extract of *Asplenium adiantum-nigrum* L. (*Aspleniaceae*)

Adrià Farràs [1,2], Víctor López [2,3], Filippo Maggi [4], Giovanni Caprioli [4], María Pilar Vinardell [1] and Montserrat Mitjans [1,*]

1 Department of Biochemistry and Physiology, Faculty of Pharmacy and Food Sciences, Universitat de Barcelona, 08028 Barcelona, Spain
2 Department of Pharmacy, Faculty of Health Sciences, Universidad San Jorge, Villanueva de Gállego, 50830 Zaragoza, Spain
3 Instituto Agroalimentario de Aragón-IA2, CITA-Universidad de Zaragoza, 50013 Zaragoza, Spain
4 School of Pharmacy, Università di Camerino, 62032 Camerino, Italy
* Correspondence: montsemitjans@ub.edu; Tel.: +34-934-024-505

**Abstract:** Ferns can be a source of polyphenolic compounds, with the fronds being the main parts described in ethnopharmacological studies. The present study screened polyphenolic phytochemicals and evaluated *in vitro* activities of a methanolic extract of *Asplenium adiantum-nigrum* L. fronds (AAM), an *Aspleniaceae* fern collected from the Prades mountains (Tarragona, Spain). Phytocharacterization by HPLC-MS/MS confirmed that the major flavonoids isolated in AAM are flavanols while the major phytochemicals are phenol acids, with chlorogenic acid being the most representative one. Cytotoxicity, cytoprotection, cellular repair activity, and phototoxicity were determined *in vitro* in the presence of 0.01, 0.1, and 1 mg/mL of the extract. No cytotoxicity was obtained in any of the cell lines tested: non-tumoral (3T3 and HaCaT) and tumoral (HeLa, HepG2, and A549) cells. Additionally, the polyphenolic extract showed greater protective effect against $H_2O_2$ in 3T3 than HaCaT cells. Despite the low total phenolic content of AAM (1405.68 mg phenolic phytochemicals/kg dry extract), the cytoprotective activity of this extract could be associated with the synergistic antioxidant action of their polyphenolic profile. In addition, the extract did not present phototoxicity against the non-cytotoxic 1.8 J/cm$^2$ dose of UVA light in both non-tumoral cell lines.

**Keywords:** *Asplenium adiantum-nigrum* L.; cytoprotection; cytotoxicity; ethnopharmacology; ferns; oxidative stress

## 1. Introduction

Living organisms have developed a set of different endogenous antioxidant mechanisms, which act by enzymatic and non-enzymatic mechanisms, with the aim of preventing reactive oxygen species (ROS) imbalances. Consequently, free radicals are indispensable for the maintenance of redox homeostasis [1]. In the last decades, oxidative stress has been associated with a sedentary lifestyle, an unhealthy diet, and continued exposure to substances with toxicological potential. Supplementation with exogenous antioxidant substances is a potentially evident strategy in the prevention and reduction of the incidence of oxidative stress, especially from plant extracts due to the synergistic antioxidant action of the phytochemicals [2].

Phenolic phytochemicals are characterized by having aromatic rings with at least one hydroxyl group. The antioxidant activity of phenolic phytochemicals are hydrogen atom transfer, single electron transfer, and transition metals chelation [3]. These phytochemicals are involved in a multitude of physiological plant functions, such as protection against oxidative stress induced by ultraviolet radiation in plants. Due to their antioxidant, anti-inflammatory, and antimicrobial action, these phytochemicals have been described as

bioactive agents in the prevention and treatment of skin diseases and other severe affections such as osteosarcoma [4,5].

Ferns are poorly represented in the pharmacopoeias compared with other vascular plants and even more so when compared with angiosperms. Multiple reasons can account for this, for example, inadequate methods of fern identification and collection [6]. However, different oriental health systems consider ferns in their texts [7–9]. In this sense, the *Pteridaceae* family have been described in Ayurveda medicine; this is the case for the species *Adiantum incisum* Forssk. (*Pteridaceae*) used for the treatment of skin disorders [7], or *Adiantum capillus-veneris* L. (*Pteridaceae*) which is described for treating measles [10]. Currently, the fern with a high number of studies as an agent in the treatment of skin conditions is the aqueous extract of *Polypodium leucotomos* (*Polypodiaceae*) fronds (PLE) because of its phenolic content [11,12].

In Europe, different properties of ferns have been described as potential nutritional agents since their fronds have been described as a source of phytochemicals with antioxidant activity [13,14]. In the Iberian Peninsula and the Balearic Islands a high biodiversity of *Pteridophytes* has been recorded [15], including different *Aspleniaceae* with veterinary ethnopharmacological uses as *Asplenium trichomanes* L. ssp. *trichomanes* and *Asplenium adiantum-nigrum* L. ssp. *onopteris* [16,17]. The Prades mountains (Tarragona, Spain) is a mountainous orography of great plant biodiversity where important representations of different species of ferns have been described, especially belonging to the *Aspleniaceae* [18]. Among them, the *Asplenium adiantum-nigrum* L. (Figure 1), also known by the scientific name *Asplenium andrewsii* A. Nelson [19], is characterized by presenting fronds with a long black petiole [20]. In Catalonia, the fronds of *A. adiantum-nigrum* L. have been reported for the treatment of chicken coccidiosis [18] and for human oral fungal infection (Catalan popular called *mal blanc*) [21].

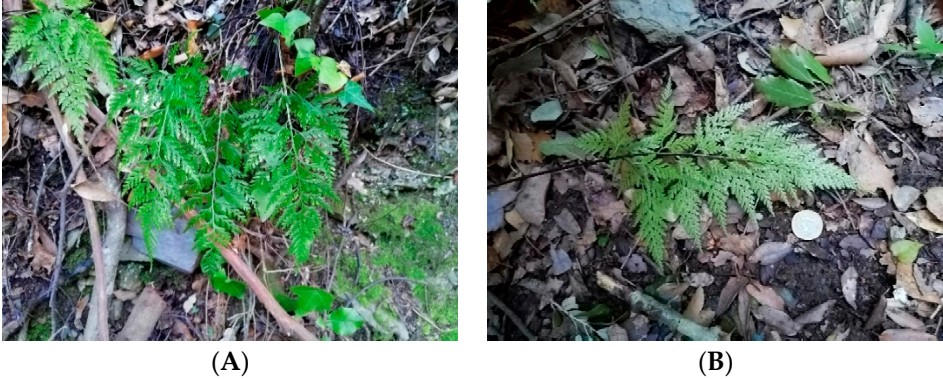

(**A**)            (**B**)

**Figure 1.** Face fronds (**A**) and underside frond (**B**) of fresh *Asplenium adiantum-nigrum* L. (*Aspleniaceae*) at the Prades mountains by Adrià Farràs. The euro coin provides a reference for the size of frond (**B**).

The extracts from the *Asplenium trichomanes* L. and *Asplenium nidus* L. (also reported as *Asplenium australasicum* (J. Sm.) Hook.) fronds, ferns from *Aspleniaceae* family, have been described for the treatment of burns (oily extract) and for wound healing (aqueous-ethanolic extract), respectively [22,23]. The main goal of our work is to characterize one of the most representative ferns of the Prades mountains (Tarragona, Spain) and to relate to their traditional and/or potential uses. As a preliminary study, we present here the characterization of a polyphenolic extract obtained from the fronds of *Asplenium adiantum-nigrum* L. analysing 38 phytochemicals. Moreover, the cytotoxic profile in various cell lines, their potential cytoprotective and cellular repair activity, phototoxic activity, and protection against ROS production were assessed in 3T3 and HaCaT.

## 2. Materials and Methods

### 2.1. Chemicals and Reagents

All reagents used were of analytical grade. Reference materials for all bioactive phytochemicals were supplied by Sigma-Aldrich (Milan, Italy), except for kaempferol-3-glucoside and quercetin that were supplied by PhytoLab (Vestenbergsgreuth, Germany). Pure reference materials were dissolved in methanol until reaching the corresponding stock solutions (1000 mg/L). These solutions were kept at 5 °C in glass-stoppered bottles until analysis. Standard working solutions were prepared fresh by diluting stock solutions with methanol (HPLC grade). Formic acid at 99% concentration and methanol were purchased from Merck (Darmstadt, Germany) and from Sigma-Aldrich (Milan, Italy), respectively. Ultrapure water (resistivity >18 MΩ cm) was obtained by filtration of deionized water with the Milli-Q SP reagent water system (Millipore, Bedford, MA, USA). All liquids were filtered through 0.2 μm polyamide filters obtained from Sartorius Stedim (Goettingen, Germany). The samples injected in the HPLC were previously filtered with Phenex™ RC 4 mm 0.2 μm syringeless filters acquired from Phenomenex (Castel Maggiore, BO, Italy). The following reagents were purchased from Sigma-Aldrich (Madrid, Spain): trypan blue (0.4%) dye, hydrogen peroxide ($H_2O_2$) 30% $w/w$, 2,5-diphenyl-3-(4,5-dimethyl-2-thiazolyl) tetrazolium bromide (MTT), dimethylsulfoxide (DMSO), 2,7-dichlorodihydrofluorescein diacetate (DCF), and chlorpromazine hydrochloride (CPZ, CAS No. 69-09-0). Dulbecco's Modified Eagle's Medium (DMEM) with and without phenol red, phosphate-buffered saline (PBS), L-glutamine solution (200 mM), trypsin-ethylenediaminetetraacetic acid (EDTA) solution (170,000 U/L trypsin and 0.2 g/L EDTA), and penicillin-streptomycin solution (10,000 U/mL penicillin and 10 mg/mL streptomycin) were obtained from Lonza (Verviers, Belgium). Thermo Scientific (Northumberland, UK) supplied HyClone fetal bovine serum (FBS). The 75 cm$^2$ culture flasks and well culture plates were purchased from TPP (Trasadingen, Switzerland).

### 2.2. Plant Material

The Prades mountains are the area where the fronds of *Asplenium adiantum-nigrum* L. have been collected. The existence in this area of this fern was previously verified by the *Banco de Datos de Biodiversidad de Cataluña* [24]. A specimen of the fresh frond was authenticated by an expert using taxonomic keys. The fronds were dried at room temperature under laboratory filter paper for 4–5 consecutive days and, finally, a dried specimen was stored at the Herbarium of Universidad San Jorge (Zaragoza, Spain): voucher no. 004-2016.

### 2.3. Preparation of Methanolic Extract of Asplenium adiantum-nigrum L. Fronds

Once the fronds were powdered, this material was macerated with methanol for 24 h at room temperature. Subsequently, the corresponding extract was filtered using a Whatman N°4 filter paper. The solvent of the corresponding filtrate was evaporated using a rotary evaporator with a thermostatic bath at 30 °C. This entire process was repeated three times to obtain the corresponding exhaustion extract [25]. Finally, the methanolic extracts were kept at −20 °C until we carried out the experiments.

### 2.4. Polyphenol Characterization of Methanolic Extract of Asplenium adiantum-nigrum L. Fronds by High Performance Liquid Chromatography-Tandem Mass Spectrometry (HPLC-MS/MS)

Phytochemical quantification was obtained by modifications of a previously described method of our group [26]. HPLC-MS/MS assay was accomplished with an Agilent 1290 Infinity series and a Triple Quadrupole 6420 purchased from Agilent Technology, located in Santa Clara, CA, USA, and connected to an electrospray ionization (ESI) source that operated in negative and positive ionization modes. The MS/MS parameters of each standard were optimized by operating flow injection analysis (FIA) by Optimizer Software (see Table S1 of Supplementary Materials). The separation of phenolic compounds was obtained by direct injection of the diluted sample (1:5) using gradient elution mode on a

Phenomenex Synergi Polar–RP C18 column (250 mm × 4.6 mm, 4 μm) using a mixture of water and methanol as solvents A and B, respectively, both with 0.1% formic acid. For column protection, a Polar RP security guard cartridge preceded the column (4 mm × 3 mm ID). The composition of the mobile phase was: 0–1 min, isocratic condition, 20% B; 1–25 min, 20–85% B; 25–26 min, isocratic condition, 85% B; and 26–32 min, 85–20% B. All solutions and solvents were filtered through a 0.2 μm polyamide filter. The injection volume was 2 μL, and the flow rate was kept at 0.8 mL/min. The temperature of the column was set to 30 °C, and the drying gas temperature in the ionization source was set to 350 °C. The flow rate of the gas was set to 12 L/min, the capillary voltage was 4000 V, and the nebulizer pressure was 55 psi. After detection in the dynamic-multiple reaction monitoring (dynamic-MRM) mode, the peak areas were integrated for quantification. The most abundant product ion was used for quantification, while the remaining ions were used for qualitative analysis. The Δ retention time (each compound's unique time window) was set at 2 min.

### 2.5. Cell Culture and Cytotoxicity Studies

Two cell lines were selected, the NIH 3T3 mouse fibroblast cell line, obtained from the repository of the European Collection of Authenticated Cell Cultures (ECACC) by purchasing them at Sigma Aldrich, and the immortalized human keratinocyte cell line HaCaT, acquired at Eucellbank (Celltec-Universitat de Barcelona, Barcelona, Spain).

Cells were grown and maintained in Dulbecco's Modified Eagle's Medium (DMEM) with 4.5 g/L glucose supplemented with 10% heat-inactivated fetal bovine serum (FBS), 2 mM L-glutamine, and 100 U/mL:100 U/mL streptomycin-penicillin mixture (10% FBS-DMEM) at 37 °C in a 5% carbon dioxide ($CO_2$)-humidified incubator. Cells were regularly checked and subsequently subcultured in 75 cm$^2$ flasks.

For cytotoxicity studies, we followed the protocol previously described in Farràs et al. 2021 [27]. Cells were treated with 0.01, 0.1, and 1 mg/mL of methanolic fronds extract of *A. adiantum-nigrum* in 5% FBS-DMEM. For each cell line and plate, a negative control was included, which corresponds to untreated cells. The cytotoxicity of AAM was finally determined by the neutral red uptake (NRU) and 2,5-diphenyl-3-(4,5-dimethyl-2-thiazolyl) tetrazolium bromide (MTT) assays after treatments, as follows.

After, the cell treatments' supernatant was extracted from each well and 100 μL of NR solution (0.05 mg/mL) or 100 μL of an MTT solution (0.5 mg/mL) in serum-free DMEM without phenol red was applied. Plates were maintained for at least 3 h at 37 °C and 5% $CO_2$ and then the supernatant was discarded. Then, in the case of NRU, 100 μL of a destain solution containing an acidic ethanol solution was added to dissolve the NR uptaken by viable cells. For the MTT, 100 μL of dimethyl sulfoxide (DMSO) was added to each well to dissolve the formazan crystals. Before reading the absorbance at 550 nm in a Tecan Sunrise microplate reader (Männedorf, Switzerland), plates were shaken for 5 to 10 min at 100 rpm/min to homogenize the well content.

*Cell viability* for NRU and MTT assays was calculated using the following equation:

$$Cell\ viability\ (\%) = \left(\frac{A_{sample}}{A_{control}}\right) \times 100$$

where $A_{control}$ and $A_{sample}$ are the absorbance of the control and each sample, respectively.

The effect of the extract over cell viability was also studied with the human tumoral cell lines: cervical cancer cell line HeLa (Eucellbank, Celltec-Universitat de Barcelona, Barcelona, Spain), liver cancer cell line HepG2 (Dr. Borràs of Experimental Toxicology and Ecotoxicology Platform (UTOX-CERETOX) of Parc Científic of Universitat de Barcelona, Barcelona, Spain), and lung cancer cell line A549 (ECACC).

### 2.6. Cytoprotective and Cellular Repair Activity of Methanolic Extract of Asplenium adiantum-nigrum L. Fronds in Non-Tumoral Cell Lines

In a similar way as previously assessed [27], hydrogen peroxide was used to induce oxidative stress. The capacity of fern extracts to prevent cellular damage or induce cellular

repair was assessed by treating cells before or after peroxide insult with 0.01, 0.1, and 1 mg/mL AAM in 5% FBS-DMEM [28]. Briefly, to prevent cellular damage, cells were pre-treated with AAM at the different concentrations for a period of 24 h. Then, the medium was discarded, cells washed with PBS, and the peroxide insult was induced by 1 mM or 2 mM $H_2O_2$ for 2.5 h. After treatment with $H_2O_2$, cell viability was determined by the NRU and the MTT assay as described above (Section 2.5).

In the case of cellular repair activity, cells were previously incubated with $H_2O_2$ for 2.5 h. Then, the medium was discarded, cells were washed and finally treated with 0.01, 0.1, and 1 mg/mL AAM in 5% FBS-DMEM for 24 h. Finally, after AAM treatment, cellular repair activity was calculated by determining cell viability with the NRU and MTT tests.

For each independent experiment and plate, correspondent negative and positive controls were included. In this case, positive controls consisted of $H_2O_2$ at 1 mM or 2 mM treated cells for 2.5 h but without pre- or post-treatment with the extract, while negative ones were untreated cells.

Cytoprotective and cellular repair activity were calculated as follows:

$$Cytoprotective\ and\ cellular\ repair\ activity\ (\%) = \left( \frac{CV_{AAM-H2O2} - CV_{H2O2}}{CV_{AAM-H2O2}} \right) \times 100$$

where *CV* is the cell viability for each condition described in the formula.

### 2.7. Phototoxicity Activity of Methanolic Extract of Asplenium adiantum-nigrum L. Fronds in 3T3 and HaCaT Cell Lines

The potential phototoxic activity of AAM was evaluated according to Farràs et al., 2021 [27], which is a modified protocol of the Organization for Economic Cooperation and Development (OECD) TG 432 (2019) [29]. In this assay, cytotoxicity of the extract was compared in the presence and in the absence of exposure to a non-cytotoxic dose of ultraviolet A light and, therefore, we can exclude the presence of phototoxic reactions.

For each experiment, two plates were prepared, one for being exposed to ultraviolet A (UVA) light and the other for remaining in the dark. Cells were treated with 0.01, 0.1, and 1 mg/mL of AAM extract solved in 0% FBS-DMEM without phenol red for 1 h and then exposed to 1.8 J/cm$^2$ UVA light or remaining in the dark. In each plate, cells not treated (negative control) and treated with chlorpromazine (37.5 µg/mL 0% FBS-DMEM without phenol red) as internal positive controls, were included. Cell viability was determined by the NRU and MTT colorimetric assays.

### 2.8. Intracellular Reactive Oxygen Species (ROS) Induced by $H_2O_2$ of Methanolic Extract of Asplenium adiantum-nigrum L. Fronds in 3T3 and HaCaT Cell Lines

The level of ROS generated in cells lines by $H_2O_2$ over a range of time was assayed pursuant to Ferreira et al. (2018) [30]. After cell pre-treatment with 0.01, 0.1, and 1 mg/mL AAM in 5% FBS-DMEM, cells were washed twice with PBS, and DCF (100 µM) was added to each well for 45 min (37 °C and 5% $CO_2$). To eliminate the excess DCF, another two PBS washes were performed and then $H_2O_2$ (1 and 2 mM) was added. Fluorescence intensity of the oxidized product of DCF was registered ($\lambda_{excitation}$ 480 nm; $\lambda_{emision}$ 530 nm) at 0, 1, 2, and 3 h by a ThermoFisher SCIENTIFIC VARIOSKAN LUX plate reader (ThermoFisher SCIENTIFIC, Waltham, MA, USA). Results were expressed as fluorescence intensity (*FI*) which have dimensionless units. The $FI_{z\ h\ vs.\ 0\ h}$ was calculated as follows:

$$Fluorescence\ Intensity_{z\ h\ Vs\ 0}\ (FI_{z\ h\ Vs\ 0\ h}) = \left( \frac{FI_{z\ h} - FI_{0\ h}}{FI_{z\ h}} \right) \times 100$$

where $FI_{z\ h}$ is the intensity fluorescence at *z* h (*z* as 1 h, 2 h, or 3 h) of incubation and $FI_{0\ h}$ the amount of fluorescence intensity at 0 h.

The *FI* for each specific time was calculated using this formula:

$$FI = \frac{Fluorence_{480 \; nm \; (excitation)}}{Fluorence_{530 \; nm \; (emision)}}$$

The $\Delta ROS$, which have dimensionless units for *FI*, was obtained using the following formula:

$$\Delta ROS_{H2O2} = \Delta ROS_{AAM \; with \; DCF-H2O2} - \Delta ROS_{DCF-H2O2}$$

*2.9. Statistical Analysis*

All experiments were executed in triplicates and almost three independent experiments were assayed, on different days, except for the cytoprotection of AAM HaCaT against 2 mM $H_2O_2$ (2.5 h) MTT for which the results correspond to $n = 2$ experiments, respectively. Statistical analysis for MTT cell viability and fluorescence intensity (FI) was performed using GraphPad Prism version 7, San Diego, CA, USA. All data were expressed as mean +/− standard error. Activities have been compared using a two-way analysis of variance (ANOVA) by Bonferroni. The results were regarded as significantly different when: $p \leq 0.05$ (*), $p \leq 0.01$ (**), $p \leq 0.001$ (***), and $p \leq 0.0001$ (****).

## 3. Results

*3.1. Phytochemical Characterization of Methanolic Extract of Asplenium adiantum-nigrum L. Fronds by High Performance Liquid Chromatography-Tandem Mass Spectrometry (HPLC-MS/MS)*

The extract contains different types of phenol species (1405.68 mg/kg dry extract), as we observe in Table 1. Moreover, the total extract presents a higher proportion of phenolic acids (81.06%) than flavonoids (18.40%). Analysis of the three main phytochemicals, chlorogenic acid (681.10 mg/kg), vanillic acid (235.19 mg/kg), and procyanidin B2 (110.16 mg/kg), also shows that phenolic acids are the major phytochemicals determined by HPLC-MS/MS.

**Table 1.** Content (mg/kg of dry extract) of 38 phenol phytochemicals in the methanolic extract of *Asplenium adiantum-nigrum* L. fronds analysed by HPLC-MS/MS ($n = 3$, RSD% ranged from 1.8 to 6.8%).

| N° | Phytochemicals | Methanolic Extract *Asplenium adiantum-nigrum* L. Fronds (AAM) |
|---|---|---|
| | **Phenolic acids** | |
| 1 | Gallic acid | 30.56 |
| 2 | Neochlorogenic acid | n.d. |
| 3 | Chlorogenic acid | 681.10 |
| 4 | p-Hydroxybenzoic acid | 109.38 |
| 5 | 3-Hydroxy benzoic acid | n.d. |
| 6 | Caffeic acid | 16.68 |
| 7 | Vanillic acid | 235.19 |
| 8 | Syringic acid | n.d. |
| 9 | p-Coumaric acid | 61.10 |
| 10 | Ferulic acid | 1.77 |
| 11 | 3,5-Dicaffeoylquinic acid | 3.62 |
| 12 | Ellagic acid | n.d. |
| | **Flavonoids** | |
| | **(A) Anthocyanins** | |
| 13 | Delphinidin-3,5-diglucoside | 2.06 |
| 14 | Delphinidin-3-galactoside | 0.52 |
| 15 | Cyanidin-3-glucoside | 64.07 |
| 16 | Petunidin-3-glucoside | n.d. |

**Table 1.** *Cont.*

| N° | Phytochemicals | Methanolic Extract *Asplenium adiantum-nigrum* L. Fronds (AAM) |
|---|---|---|
| 17 | Pelargonidin-3-rutinoside | n.d. |
| 18 | Pelargonidin-3-glucoside | n.d. |
| 19 | Malvidin-3-galactoside | n.d. |
| | **(B) Flavonols** | |
| 20 | Rutin | 5.54 |
| 21 | Isoquercitrin | 3.17 |
| 22 | Quercitrin | 0.99 |
| 23 | Myricetin | 0.71 |
| 24 | Kaempferol-3-glucoside | 36.85 |
| 25 | Quercetin | 0.74 |
| 26 | Isorhamnetin | 0.18 |
| 27 | Hyperoside | 5.59 |
| 28 | Kaempferol | 1.88 |
| | **(C) Flavan-3-ols (Flavanols)** | |
| 29 | Catechin | n.d. |
| 30 | Epicatechin | 7.69 |
| 31 | Procyanidin B2 | 110.16 |
| 32 | Procyanidin A2 | 3.63 |
| | **(D) Dihydrochalcones** | |
| 33 | Phloridzin | n.d. |
| 34 | Phloretin | n.d. |
| | **(E) Flavanones** | |
| 35 | Hesperidin | 14.81 |
| 36 | Naringin | n.d. |
| | **Stilbenes** | |
| 37 | Resveratrol | n.d. |
| | **Non-phenolic acids** | |
| 38 | Trans-cinnamic acid | 7.69 |
| | Total phenol content | 1405.68 |

nd = not detected.

Other phenol acids as neochlorogenic acid, 3-hydroxybenzoic acid, syringic acid, and ellagic acid were not detected. Interestingly, we determined a variability of flavonoids in residual amounts, although they represent a low proportion of the total phytochemicals determined. Among them are quercitrin, quercetin, myricetin, delphinidin-3-galactoside, and isorhamnetin.

### 3.2. In Vitro Cell Assays

Determination of cell viability by the NRU assay shows no significant differences between treated and untreated cells independently of the cell line studied (data not shown). Thus, according to NRU data, the extract is not cytotoxic in the conditions assayed. For this reason, only data obtained by the MTT assay is analysed.

3.2.1. Cytotoxicity of Methanolic Extract of *Asplenium adiantum-nigrum* L. Fronds in Non-Tumoral and Tumoral Cell Lines

We evaluated the effect of increasing concentrations of AAM at 0.01, 0.1, and 1 mg/mL, in non-tumoral (3T3 and HaCaT) and tumoral (HeLa, HepG2 and A549) cell lines. As observed in Figure 2, AAM does not induce an important cytotoxicity in 3T3 and HaCaT (Figure 2A,B, respectively). However, a slightly lower viability has been recorded for fibroblasts than for keratinocytes; being the lowest cell viability recorded at 0.1 mg/mL AAM in 3T3 (62.4% cell viability).

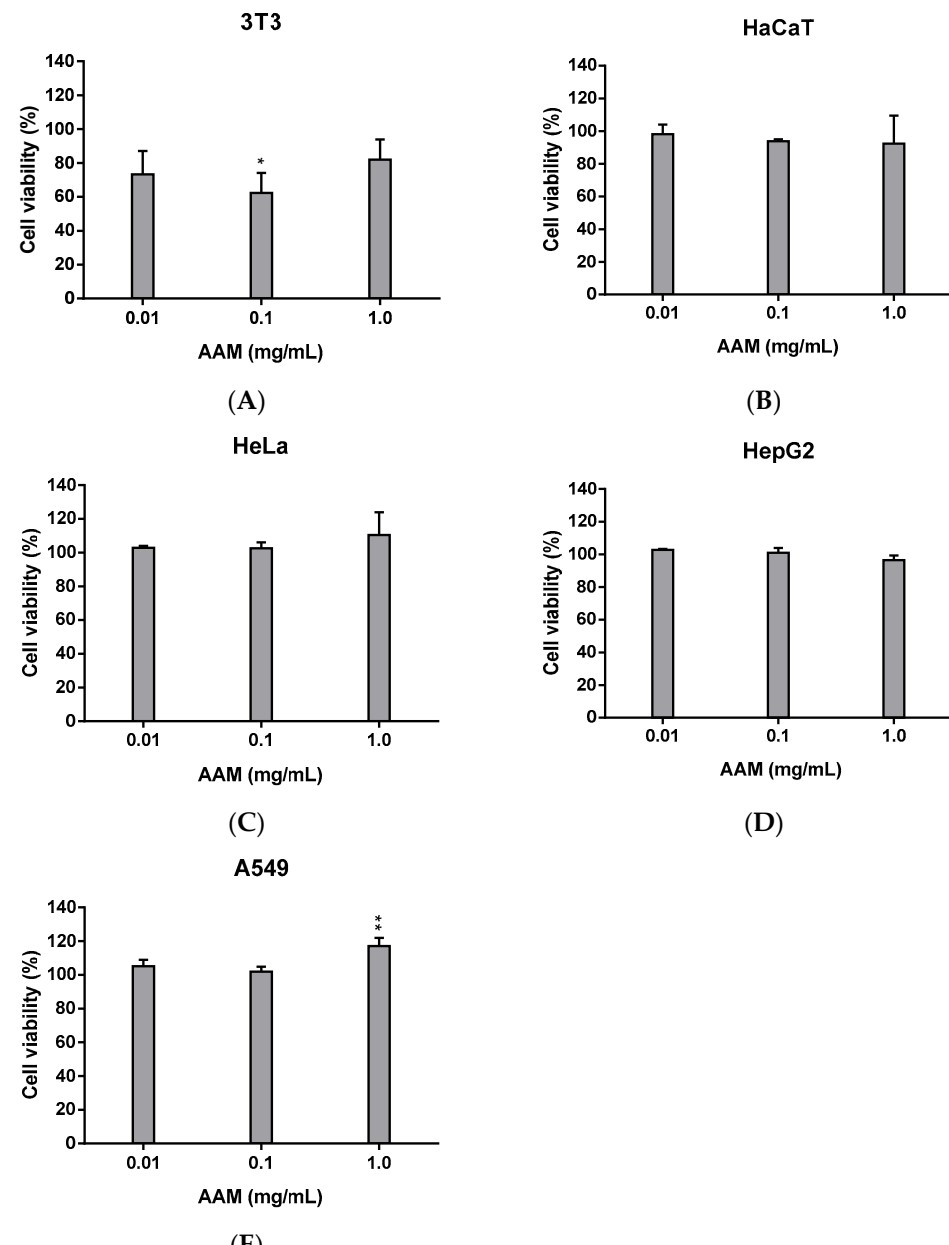

**Figure 2.** Cytotoxic activity of AAM in 3T3 (**A**), HaCaT (**B**), HeLa (**C**), HepG2 (**D**), and A549 (**E**) cell lines obtained by MTT assay and expressed as percentage of cell viability with respect to control cells. Results are expressed as mean $\pm$ standard error of $n$ = 3. Control or untreated cells were maintained with culture medium. A two-way analysis of variance (ANOVA) and a Bonferroni post hoc assay have been performed. Statistical differences were considered as follows: * $p \leq 0.05$, and ** $p \leq 0.01$ in comparison with untreated cells (negative control).

For the tumoral cell lines (Figure 2C–E), we failed to find any cytotoxic behaviour at the conditions tested for the extract. For this reason, the following studies were only performed with non-tumoral cell lines.

### 3.2.2. Cytoprotective Activity of Methanolic Extract of *Asplenium adiantum-nigrum* L. Fronds in 3T3 and HaCaT Cell Lines

We evaluated the capacity of the AAM extract to prevent oxidative damage induced by $H_2O_2$ (2 and 1 mM $H_2O_2$ for 2.5 h). However, we only found some cytoprotective activity against 2 mM $H_2O_2$ for 2.5 h (results obtained for 1 mM $H_2O_2$ are presented in Figure S1 of Supplementary Materials).

Figure 3 shows cell viability obtained for both cell lines, 3T3 (Figure 3A) and HaCaT (Figure 3B). Cell viability of positive controls (cells treated by $H_2O_2$ but not with AAM extract) indicate that 3T3 cells seem to be more sensitive to $H_2O_2$ than HaCaT (32.6% and 45.9% cell viability, respectively), which could explain in part the higher cytoprotection observed in 3T3. However, it is only in the case of 3T3 treated with 1 mg/mL AAM that we found statistically significant cytoprotective activity, with a calculated value of cytoprotective effect greater than 30% (Table 2). According to Siddiqui, et al., 2011, cellular repair activity (i.e., post-treatment by the extract) presents good results only in the case that protective effects have been observed in the pre-treatment, since repairing mechanisms require more cellular energy than the protective ones. Consequently, cellular repair activity was only assessed in 3T3 cells.

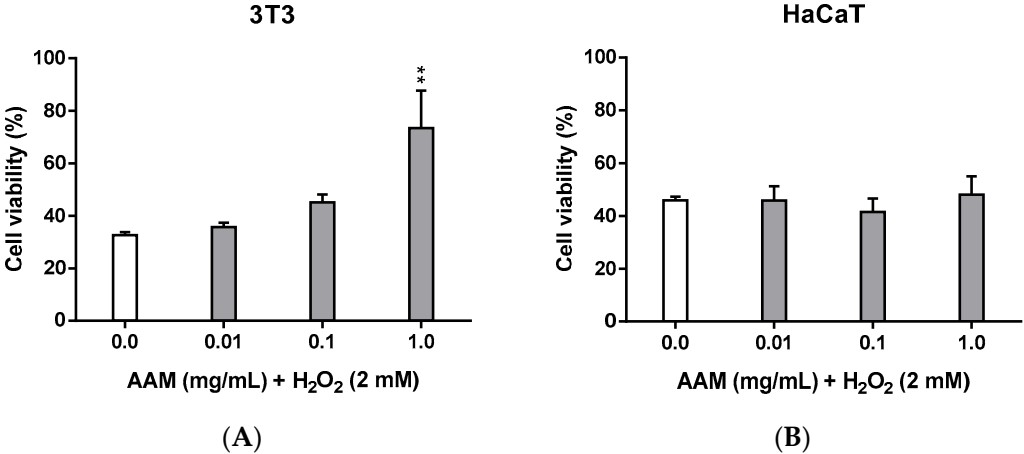

(**A**)                                                    (**B**)

**Figure 3.** Cytoprotective activity of AAM in 3T3 (**A**) and HaCaT (**B**). Cells were incubated 24 h in the absence (white column) or in the presence of AAM (gray columns) and subsequently treated with 2 mM $H_2O_2$ for 2.5 h and finally cell viability was determined by MTT assay and expressed as a percentage with respect to untreated cells. $H_2O_2$ cell viability in absence of the extract was used as a positive control. Results are expressed as mean $\pm$ standard error of $n = 3$ (3T3) and $n = 2$ (HaCaT). A two-way analysis of variance (ANOVA) and a Bonferroni post hoc assay have been performed. Statistical differences were considered as follows: ** $p \leq 0.01$ in comparison with positive control.

**Table 2.** Cytoprotective activity of AAM on 3T3 and HaCaT cells.

| Concentration of AAM (mg/mL) | 0.01 | 0.1 | 1 |
|---|---|---|---|
| Cytoprotection activity (%) [a] in 3T3 | 8.7% | 27.7% | 55.6% |
| Cytoprotection activity (%) [a] in HaCaT | 0.0% | 0.0% | 4.4% |

[a] Percentage of cytoprotection activity has been obtained from the following relation $[(CV_{AAM-H2O2} - CV_{H2O2})/CV_{AAM-H2O2}] \times 100$.

### 3.2.3. Cellular Repair Activity of Methanolic Extract of *Asplenium adiantum-nigrum* L. Fronds in 3T3 Tissue Cell Line

In a similar way as in the previous experiments of cytoprotection, 3T3 cells suffer a high decrease in viability (21.9%) 24 h after being treated with 2 mM $H_2O_2$ for 2.5 h (Figure 4).

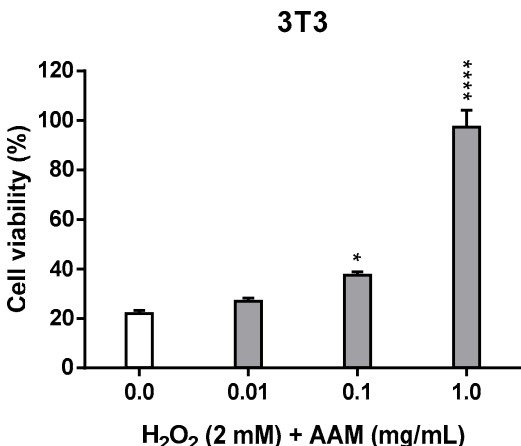

**Figure 4.** Cellular repair activity of AAM in 3T3 cells. Cells previously treated with 2 mM $H_2O_2$ for 2.5 h were incubated in the absence (white column) or in the presence of AAM for 24 h (gray columns) and finally cell viability was determined by MTT assay and expressed as a percentage with respect to untreated cells. $H_2O_2$ cell viability in absence of the extract was used as a positive control. Results are expressed as mean $\pm$ standard error of $n = 3$. A two-way analysis of variance (ANOVA) and a Bonferroni post hoc assay have been performed. Statistical differences were considered as follows: * $p \leq 0.05$, and **** $p \leq 0.0001$ in comparison with positive control.

Results obtained here indicate a tendency to increase viability in a concentration dependent manner (Table 3). However, statistical significance of such increase is only recorded at 0.1 and 1 mg/mL extract.

**Table 3.** Cellular repair activity of AAM in 3T3 cell line for 2 mM $H_2O_2$ during 2.5 h by MTT assay.

| Concentration of AAM (mg/mL) | 0.01 | 0.1 | 1 |
|---|---|---|---|
| Cellular repair activity (%) [a] | 18.8% | 41.6% | 77.5% |

[a] Percentage of cellular repair activity has been obtained from the following relation [($CV_{\text{AAM-H2O2}} - CV_{\text{H2O2}})/CV_{\text{AAM-H2O2}}] \times 100$.

### 3.2.4. Phototoxicity Activity of Methanolic Extract of *Asplenium adiantum-nigrum* L. Fronds in 3T3 and HaCaT Tissue Cell Lines

UVA light sensitivity for 3T3 and HaCaT cells was determined in each experiment by including negative controls (cells not treated) and comparing their cell viability in irradiated and dark conditions. In our case, 3T3 showed a cell viability of 63.3% and HaCaT 75.0%, indicating moderate sensitivity of our cells to UVA light, a fact that could account to underestimate the phototoxic behaviour of our products. Nevertheless, cells treated with CPZ, a well-known phototoxic chemical (internal positive control), show a high decrease in cell viability when exposed to light, which is more pronounced in the case of HaCaT cells, indicating that phototoxicity has been induced.

Figure 5 presents the viability for 3T3 and HaCaT cells obtained by MTT in dark and UVA light conditions. In both cases, cell viability of cells treated with AAM is not affected by UVA light exposition and thus phototoxicity activity of the extract can be discarded. Interestingly, cells treated by the AAM extract show a high increase of cell viability, which is independent of UVA treatment, achieving values of almost 200% in the case of 3T3 in some conditions, although no statistical signification is found.

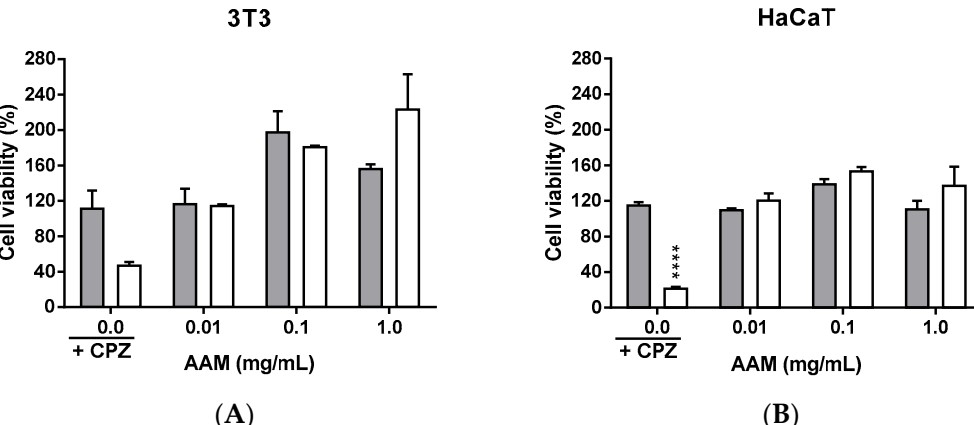

(A)  (B)

**Figure 5.** Phototoxicity activity of AAM in 3T3 (**A**) and HaCaT (**B**) cell lines obtained by MTT assay and expressed as percentage of cell viability respect to the correspondent control cells. Chlorpromazine cell viability was used as positive control. Gray columns correspond to non-exposed cells to UVA light and white columns correspond to cells exposed to $1.8 \, J/cm^2$ of UVA light. Results are expressed as mean $\pm$ standard error of $n = 3$. A two-way analysis of variance (ANOVA) and a Bonferroni post hoc assay have been performed. Statistical differences were considered as follows: **** $p \leq 0.0001$ in comparison with the equivalent non-irradiated condition homologue.

3.2.5. Intracellular ROS Induced by $H_2O_2$ of Methanolic Extract of *Asplenium adiantum-nigrum* L. Fronds in 3T3 and HaCaT Cell Lines

DCF probe was used to determine the production of intracellular ROS expressed as FI as shown in Figure 6 for 2 h. Absolute ROS values increase with time (1, 2, and 3 h), but no differences in pattern behaviour is observed (data not shown). Thus, we only present here the ROS production at 2 h.

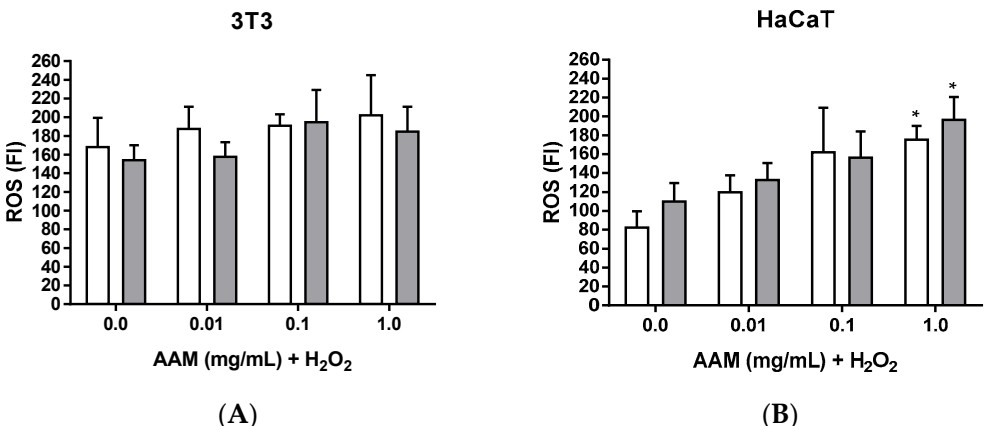

(A)  (B)

**Figure 6.** Intracellular ROS induced by 1 and 2 mM $H_2O_2$ for 2 h treatment after 24 h incubation with AAM in 3T3 (**A**) and HaCaT (**B**) cells. $H_2O_2$ treated cells were used as positive control. White columns correspond to 1 mM $H_2O_2$ and gray columns correspond to 2 mM $H_2O_2$. Results are expressed as mean $\pm$ standard error of $n = 3$. A two-way analysis of variance (ANOVA) and a Bonferroni post hoc assay have been performed. Statistical differences were considered as follows: * $p \leq 0.05$ in comparison with the corresponding positive control.

ROS production shows a similar pattern for 1 mM and 2 mM $H_2O_2$ with a slight tendency to increase in parallel to AAM extract concentration but the only statistically significant case is that of HaCaT at 1 mg/mL of AAM (Figure 6B). This observation can be attributed to the lower values of ROS production induced by $H_2O_2$ in the case of HaCaT.

Table 4 summarizes the different increases of ROS production in the various conditions assessed. It is observed that in the case of HaCaT the amount of ROS has a significant rise as stated before.

**Table 4.** Intracellular $\Delta$ROS [a] induced by 1 and 2 mM $H_2O_2$ for 2 h respect 0 h at different concentrations of AAM in 3T3 and HaCaT.

| Cell Line | 3T3 | | | HaCaT | | |
|---|---|---|---|---|---|---|
| Concentration of AAM (mg/mL) | 0.01 | 0.1 | 1 | 0.01 | 0.1 | 1 |
| 1 mM $H_2O_2$ | 19.6 | 22.7 | 34.0 | 37.5 | 79.7 | 93.1 |
| 2 mM $H_2O_2$ | 3.8 | 40.8 | 30.6 | 22.8 | 46.4 | 86.4 |

[a] Intracellular $\Delta$ROS cytoprotective activity has been obtained from the following relation $ROS_{AAM\ with\ DCF-H2O2}$—$ROS_{DCF-H2O2}$ expressed as fluorescence intensity (*FI*).

The increase of ROS generation should be studied as the extract confers some cytoprotective effect to 3T3 when treated with 2 mM $H_2O_2$ for 2.5 h but not to HaCaT (Figure 3 and Table 2).

## 4. Discussion

Many of the active principles used in the treatment of diseases have emerged from ethnomedicinal studies. The correct documentation of medicinal uses is essential before they fall into oblivion [16]. For example, in an ethnopharmacological study of medicinal plant uses in Pakistan, the treatment of measles is documented with an extract of the fronds of *A. capillus-veneris* (*Pteridaceae*) [10].

To date, different studies have reported antioxidant activity of different fern species [9,31,32], with the PLE being the most studied fern for its antioxidant properties that have been considered useful for the treatment of skin disorders [27,33,34]. The phytochemicals described in PLE are phenolic acids belonging to the hydroxycinnamic acid (*p*-coumaric acid, caffeic acid, chlorogenic acid, and ferulic acid) and benzoic acid (vanillic acid) [35]. In a similar way, our research group determined the polyphenolic profile of the methanolic extract of *Polypodium vulgare* L. fronds (PVM), another fern of the *Polypodiaceae* family, by HPLC-DAD. Among the high number of phenolic acids identified, 3-O-caffeoylquinic acid or chlorogenic acid (hydroxycinnamic acid), a potent antioxidant used in the treatment of oxidative stress and cancer [5], was reported as one of the major constituents [27]. Taken together, these studies indicate that the fronds of *Polypodiaceae* are important reservoirs of polyphenols and related compounds. Therefore, ferns are considered important sources of phytochemical diversity, such as flavonoids [32,36]. The *Aspleniaceae* family, besides being a source of phenolic acids, is also regarded as a source of flavonols, one of the types of flavonoids [37]. In our case, HPLC-MS/MS determination indicates that flavanols are the major flavonoids isolated from AAM, with procyanidin B2 (flavanol) the main flavonoid isolated. Procyanidins are oligomers of catechins that share the structure of flavanols. Catechins and the phytochemicals related to them act directly in the prevention of oxidative stress induced by solar radiation on the skin [38]. Moreover, the described potent antioxidant activity of the flavonoids catechin, epicatechin, and kaempferol is concluded to be the trigger of the chemopreventive activity in melanoma and lung cancer, respectively [39]. In our case, although flavanols are the largest flavonoids in the extract (8.64% total phytochemicals determined), we also find flavonols, which represent 3.96% of total extract and whose main representative is kaempferol-3-glucoside.

Methanol solvent is frequently used to extract flavonoids from fern fronds such as *Lygodium venustum* Sw. (*Lygodiaceae*) [40] and *Asplenium normale* D. Don (*Aspleniaceae*) [41]. However, due to the cytotoxicity of methanol, it must be removed from the extract; otherwise, it could give false positives in cytotoxicity tests [42]. Currently, *in vitro* cytotoxicity tests are the first test to evaluate the biocompatibility of any plant drug for its subsequent use in health-related products [40]. However, one limiting factor for *in vitro* tests is the solubility and penetration of the extract in the target of the structure, which determines

the cellular viability [43]. To avoid that, AAM requires previous ultrasonic agitation before being applied to the cells. Another factor to bear in mind is the bioassay chosen to evaluate the cytotoxic behaviour of phytochemicals. In our case, we evaluated the potential cytotoxicity of AAM by the NRU assay (cellular viability determined by the integrity of the lysosome membrane) and the MTT assay (cellular viability obtained by mitochondrial metabolic activity). Our results indicate that MTT is more sensitive than NRU, and that the extract penetrates the cell and consequently reaches the mitochondria localized in the cytosol [44–46]. However, MTT assay has some disadvantages that are dependent on the cell ability to overcome cell death. For example, the damaged mitochondria may be still able to reduce the tetrazolium salt. To avoid these false positives of the MTT assay, a periodic control of cell structure and density should be carried out [47,48]. The reduced number of cytotoxicity studies together with the complex comparison of bioactivities of fern extracts makes it difficult to compare our results *in vitro* with other studies. Differences in extraction solvent polarity, inconsistency on extract concentrations tested, and numerous cellular viability assays performed among the different studies are some variables that make a reliable comparison difficult.

3T3 is considered a widely used cell line to determine the doses to be tested in subsequent systemic toxicological studies *in vivo* [49]. In cytotoxicity assays, the use of a minimum of two cell lines with different cell origins is recommended. Moreover, the use of 3T3 and HaCaT is frequently combined when characterizing phytochemicals with dermal applications [50,51]. Here, AAM extract induces low cytotoxic effects according to cell viability values obtained with MTT. In a similar way, no cytotoxicity on HaCaT was described for the 2-propanol extract from the fronds of *Ophioglossum vulgatum* L. (*Ophioglossaceae*) and demonstrated healing properties. This extract contains flavonols derived from quercetin and kaempferol [52]. The AAM extract, as the *O. vulgatum* extract, has a representative amount of flavonols, among which are quercetin and kaempferol. However, if in future studies the healing activity of the AAM extract is determined, the flavanols will probably be responsible for this bioactivity as they are the main flavonoids of this extract.

The gallate group, structure of gallic acids, is a powerful antioxidant. However, on certain occasions it alters crucial cellular functions, resulting in cytotoxicity [53]. For example, green tea is a source of catechins and catechin derivatives with the presence of the gallate group catechin gallate. Catechin, as well as its derivatives such as epicatechin, have been shown to have antiproliferative effects in different carcinoma cell lines. In addition, beneficial effects of green tea application on smokers' preneoplastic oral cavity lesions has also been reported [54]. For this reason, it is required to evaluate the cytotoxicity of phytochemicals containing the gallate group in different tumoral and non-tumoral cell lines *in vitro* as a previous step of its potential uses. In our case, the absence or low cytotoxic activity of AAM obtained by the MTT assay in the non-tumoral and tumoral cell lines, rules out the potential cytotoxicity a priori conceived by the content of gallic acid and phytochemicals with gallate group.

Other three ferns from the *Aspleniaceae* family (*Asplenium ceterach* L., *Asplenium scolopendrium* L., and *Asplenium trichomanes* L.) also did not show any cytotoxicity on the lung cancer cell line A549, as described by Petkov, et al. (2021) [55]. However, in the same study, these ferns show a selective cytotoxicity for HeLa cells in contrast to our results. One explanation can be derived from the different phytochemical profile described for these three ferns because they contain flavonols like the AAM extract, but also fatty acids. Petkov, et al. (2021) [55] suggest that their fatty acids composition may contribute to the selective antiproliferative effects on HeLa cells.

Currently, the PLE extract has been described as a powerful chemopreventive agent because of the powerful antioxidant action of its polyphenolic acids. Among these polyphenolic acids, we find different derivatives of cinnamic acid and benzoic acid, such as chlorogenic acid and vanillic acid, respectively. However, derivatives of benzoic acids have a lower antioxidant capacity compared with cinnamic acids because the first group lacks

the -CH=CH-COOH chain, which is the structure that provides the electron donor properties [35]. The main phytochemical and most representative of the big number of phytochemicals determined in AAM by HPLC-MS/MS is chlorogenic acid.

3T3 cells have shown greater sensitivity than HaCaT against AAM in the cytotoxicity and the cytoprotective assays performed. In different studies with flavanol fractions, a greater sensitivity has also been obtained in 3T3 than in HaCaT [50]. The positive controls of the cytoprotection assays have also presented a higher sensitivity in the 3T3 line than in the HaCaT, agreeing with Maier, et al. (1991) [56] who described higher sensitivity of 3T3 to oxidizing and irritating agents than HaCaT. The statistically significant antioxidant *in vitro* protection effects of AAM in the 3T3 cell line confirm our previous studies that indicated good antioxidant capacity determined by the *in chemico* models DPPH, ORAC, and superoxide radicals generated by xanthine/xanthine oxidase [25]. However, we failed to demonstrate this fact in HaCaT cells. Differences between the two cell lines (origin, source, metabolic capacity, among other variables) could explain our findings. When oxidative stress is generated before potential antioxidant treatment, greater cell damage is observed (Figure 4). This fact generally entails a greater cytoprotective effect in the pre-treatment than in the post-treatment, as reported in the case of the *in vitro* protective capacity of the curcumin (curcuminoid) in PC12 (a rat pheochromocytoma cell line) against $H_2O_2$ [57]. According to outcomes in the cytoprotective assay, the 3T3 cell line was selected for the cellular repair one. For both assays, the protective effects observed were directly related to the concentration of AAM assayed. The greater protection induced by AAM in 3T3 in front $H_2O_2$ injury in post-treatment than in pre-treatment suggests the need to evaluate the protective mechanisms of this extract. However, open questions remain, and further investigations are still needed as a consequence of the multi-target mechanisms of extracts derived of their phytochemicals content [58].

The phototoxicity assay OECD TG 432 [29], based on the determination of phototoxic substances using the NRU and BALB/c 3T3 cell line, has high sensitivity to identify photosensitive substances failing somehow in some cases and thus reporting false positive results. For this reason, using another cell line, especially a cutaneous cell line, can contribute to minimize the overestimation of phototoxicity reactions. We chose HaCaT cells to be a human keratinocyte cell line and thus complementary to the fibroblast cell line 3T3. The different location and consequently the different function of these cell lines in skin is an adequate combination for analysing phototoxicity [59]. UVA produces different alterations at the cellular level, including disorganization of the cytoskeleton. In a similar way as reported for PLE [60], AAM has not induced phototoxicity under our conditions of UVA irradiation in both cell lines studied. Surprisingly, we observe an increase in cell viability both in non-irradiated and irradiated cells treated with the extract, which is more evident in the case of 3T3 at 0.1 and 1 mg/mL. Comparing these results with those of cytotoxicity, some questions emerge. Experimental conditions of time exposition, media, and temperature can explain such different behaviour. However, further studies at short time exposition could clarify the open questions encountered here. On the other hand, the main flavonoid identified in AAM is procyanidin B2, a key phytochemical in the protection of skin cells exposed to UVB by reducing lipid peroxidation [61].

Mitochondria is the basic structure to generate energy in aerobic organisms, and thus also generating ROS [62]. In physiological situations, endogenous antioxidant systems counteract ROS. However, on certain occasions, the decrease of ROS may be attributed to a decrease of cellular viability, as dead cells no longer generate ROS. This fact may explain the minor ROS production observed for the 3T3 positive control at 2 mM $H_2O_2$ compared with the corresponding one at 1 mM $H_2O_2$. Nevertheless, it is necessary to supplement this assumption with cell count assays. It is well known that flavonoids, due to their radical scavenging by an ortho-dihydroxy (catechol) structure in the B ring, 2,3-double bond in conjugation with a 4-oxo function in the C ring and hydroxyl groups at positions 3 and 5 [63–65], have a greater antioxidant capacity than phenolic acids [3]. It is possible that, due to the powerful antioxidant action of our extract on ROS production, a prooxidant effect

could have been generated, and thus an increase in ROS with respect to the corresponding positive controls. Phytochemicals can present different antioxidant behaviour depending on the global phytochemicals of the extract due to synergistic antioxidant actions [66,67], as is the case with the synergistic antioxidant action of (−)-epicatechin and its galloylated derivatives in contact with the flavonols quercetin, kaempferol, and myricetin described for water extracts of white tea [68]. Recently, it has been described that flavanols are the main flavonoids in AAM together with a large amount of mangiferin and mangiferin glucoside, which are both xanthones. Therefore, Zivkovic, et al. (2020) attributes the *in vitro* antioxidant activity of AAM to the presence of these xanthones and, consequently, xanthones are regarded as polyphenolic phytochemicals with high antioxidant activity *in vitro* [69]. In our investigations, we also determined that flavanols are the main flavonoids of AAM, however, we have not analysed the presence or absence of xanthones in the HPLC-MS/MS. This fact leads us to speculate that the observed antioxidant activity of the AAM extract in the cell lines evaluated is a consequence of the synergistic antioxidant action of the flavanols with the xanthones, a fact that characterizes AAM as a valuable reservoir of polyphenols.

## 5. Conclusions

Ferns are considered reservoirs of interesting phytochemicals because of their adaptation to the environment. Many of them are potential candidates for ethnopharmacological studies to contribute to the assessment and conservation of plant biodiversity. Methanolic extracts of *Aspleniaceae adiantum-nigrum* L. fronds are composed by a variety of polyphenols as phenol acids (as chlorogenic acid) and flavanols (as procyanidin B2), which confer the biological activities reported here for the extract. Thus, the cellular repair capacity observed in the 3T3 cells treated with AAM after peroxide insult can be explained by the antioxidant capacity reported in previous studies. In addition, AAM has shown no relevant cytotoxicity in the five cell lines and no phototoxic adverse effects when treated fibroblasts and keratinocytes were exposed to a non-cytotoxic dose of UVA light. These observations, together with its capacity to inhibit tyrosinase enzyme, open the possibility of using AAM in cosmetic formulations as ingredients in sunscreens or hyperpigmentation treatments, but also in pharmaceutical preparations for wound healing. Therefore, the cellular mechanisms and specific phytochemicals involved in such activities should be further investigated.

**Supplementary Materials:** The following supporting information can be downloaded at: https://www.mdpi.com/article/10.3390/horticulturae8090815/s1, Figure S1: Cytoprotective activity of AAM in 3T3 (**A**) and HaCaT (**B**) cell lines for 1 mM $H_2O_2$ during 2.5 h by MTT assay and expressed as percentage of cell viability respect to untreated cells; Table S1: HPLC–MS/MS acquisition parameters (dynamic-MRM mode) used for the analysis of the 38 marker compounds.

**Author Contributions:** Conceptualization M.M. and V.L.; methodology, A.F., M.M. and F.M.; analysis, A.F., F.M., G.C. and M.M.; investigation, A.F., M.M. and V.L.; resources, M.M., M.P.V. and V.L.; writing—original draft preparation, A.F.; writing—review and editing, M.M. and V.L.; supervision, M.M. and V.L.; funding acquisition M.M. and M.P.V. All authors have read and agreed to the published version of the manuscript.

**Funding:** This research was financially supported by project 307,629 of Fundació Bosch & Gimpera—Universitat de Barcelona.

**Acknowledgments:** We acknowledge the technical support of Universitat de Barcelona and Universidad San Jorge.

**Conflicts of Interest:** The authors declare no conflict of interest.

## Abbreviations

| | |
|---|---|
| **3T3 = NIH 3T3** | NIH 3T3 mouse fibroblast cell line |
| **A549** | Human Caucasian lung carcinoma |
| **AAM** | Methanolic extract of *Asplenium adiantum-nigrum* L. fronds |
| **CO₂** | Carbon dioxide |
| **CPZ** | Chlorpromazine hydrochloride |
| **CV** | Cell Viability |
| **E** | Ultraviolet dose |
| **DCF** | 2,7-dichlorodihydrofluorescein diacetate |
| **DMEM** | Dulbecco's Modified Eagle's Medium |
| **DMSO** | Dimethyl sulfoxide |
| **ECACC** | European Collection of Authenticated Cell Cultures |
| **EDTA** | Ethylenediaminetetraacetic acid |
| **DPPH** | 2,2-diphenyl-1-picrylhydrazyl |
| **FBS** | Fetal Bovine Serum |
| **FI** | Fluorescence Intensity |
| **HaCaT** | Spontaneously immortalized human keratinocyte cell line |
| **HeLa** | Human cervix epitheloid carcinoma |
| **HepG2** | Human Caucasian hepatocyte carcinoma |
| **HPLC-MS/MS** | High performance liquid chromatography-tandem mass spectrometry |
| **H₂O₂** | Hydrogen peroxide |
| **MTT** | 2,5-diphenyl-3-(4,5-dimethyl-2-thiazolyl) tetrazolium bromide |
| **NR** | Neutral Red |
| **NRU** | Neutral Red Uptake |
| **OECD** | Organisation for Economic Cooperation and Development |
| **ORAC** | Oxygen Radical Absorbance Capacity |
| **PBS** | Phosphate Buffered Saline |
| **PLE** | Aqueous extract of *Polypodium leucotomos* fronds |
| **PVM** | Methanolic extract of *Polypodium vulgare* L. fronds |
| **ROS** | Reactive Oxygen Species |
| **UV** | Ultraviolet |
| **UVA** | Ultraviolet A |
| **WHO** | World Health Organization |

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
