# Peer review of "Chemical Composition and Cytoprotective Activities of Methanolic Extract of Asplenium adiantum-nigrum L. (Aspleniaceae)"

_horticulturae, doi:10.3390/horticulturae8090815_

Round 1
Reviewer 1 Report
This study investigates the chemical compositions and cytoprotective activities of methanolic extract of Asplenium adiantum nigrum L. (Aspleniaceae). HPLC-MS/MS was used to identify the polyphenols in the extracts, and non-tumoral (3T3 and HaCaT) and tumoral (HeLa, HepG2 and A549) cells were used to conduct the cytoprotection assays. Overall, the manuscript was well organized and wroten and its data are sufficient. I agree with the publication of this article. But before publication, the authors need to consider the following comments.
1. The title can be improved. A good title should make the main content of the article clear at a glance. “Chemical compositions and cytoprotective activities of methanolic extract of Asplenium adiantum nigrum L. (Aspleniaceae)” can be considered.
2. The abstract should be rewritten to give more data.
3. The introduction is too long and does not address the scientific issues clearly. The authors can use the following structure:
1) Introduction oxidative stress (2-3 sentences).
2) Polypehnols are considered as powerful cytoprotective agents against oxidative stress (1-2 sentences).
3) Mechanisms of cytoprotective effects of polypehnols (1-2 sentences).
4) Introduction ferns, and the polyphneols of ferns and their cytoprotective effects (2-3 sentences).
5) Significance and aim of current research (2-3 sentences).
4. Line 62: ... a therapeutic tool for the prevention and treatment of diseases.
5. Line 167-171: I'm not sure it's okay to have two languages in the same article.
6. Section 2.3: The detection wavelength of the HPLC system is? and the quantitative analysis is incorrect.
7. Section 3.1: The retention times and MS data of each molecule should be presented in the manuscript. The chemical compositions of polyphneols from the plant should be presented as a main part.
8. Line 438-439: The sentence makes no sense.
9. Line 477-479: Change to “Petkov et al. (2021) [58] suggest that their fatty acids composition may contribute to the selective antiproliferative effects on HeLa cells.”
Section 4: The conclusion part is poorly written and very short. One must get the clear understanding of your work by reading your conclusion and your’s conclusion part is weak, improve it.
Author Response
Dear reviewer,
Thanks for your time and comments. Please find below a point-by-point our answers to your comments. In the revised manuscript, additions and corrections are highlighted in blue colour.
- The title can be improved. A good title should make the main content of the article clear at a glance. “Chemical compositions and cytoprotective activities of methanolic extract of Asplenium adiantum nigrum L. (Aspleniaceae)” can be considered.
Answer: We have considered the proposed title and we have changed it accordingly.
- The abstract should be rewritten to give more data.
Answer: We have rewritten the abstract.
- The introduction is too long and does not address the scientific issues clearly. The authors can use the following structure:
Answer: As requested by the reviewer the introduction was shortened and now the extension is half of the previous one.
- Line 62: ... a therapeutic tool for the prevention and treatment of diseases.
Answer: Done.
- Line 167-171: I'm not sure it's okay to have two languages in the same article.
Answer: We apologise for the mistake; we have rewritten the text in the correct language.
- Section 2.3: The detection wavelength of the HPLC system is? and the quantitative analysis is incorrect.
Answer: The method is HPCL-MS/MS so we did not use a certain wavelength as the detection system is based on mass spectrometry. This method is validated and has already been published in: https://www.sciencedirect.com/science/article/abs/pii/S0308814621017490?via%3Dihub
The wavelength is not available as we haven't analyzed samples with DAD (diode array detector), but with a MS/MS triple quadrupole system.
- Section 3.1: The retention times and MS data of each molecule should be presented in the manuscript. The chemical compositions of polyphneols from the plant should be presented as a main part.
Answer: Please find enclosed a Table with the HPLC–MS/MS acquisition parameters (dynamic-MRM mode in MS/MS) used for the analysis of the 38 marker compounds that includes the retention time and all MS data. This table will be added as supplementary material of the manuscript.
- Line 438-439: The sentence makes no sense.
Answer: The sentence has been rewritten.
- Line 477-479: Change to “Petkov et al. (2021) [58] suggest that their fatty acids composition may contribute to the selective antiproliferative effects on HeLa cells.”
Answer: The sentence has been rewritten.
- Section 4: The conclusion part is poorly written and very short. One must get the clear understanding of your work by reading your conclusion and your’s conclusion part is weak, improve it.
Answer: As suggested by the reviewer the conclusion was extended and improved.
We want to thank you the reviewer for its constructive criticisms and suggestions.
Reviewer 2 Report
This study presents a phytochemical characterization of the extract of Asplenium adiantum-nigrum L. (AAM) and investigated its capacity to protect cells from an oxidative in vitro environment. All the experiments that tested the potential cytoprotective role of AAM were based on viability/proliferation assays (MTT assays) performed in two different cell culture models of oxidative stress (OS), one induced by H2O2 and the other induced by UV exposure.
1. Perhaps the major criticism I have relates to the H2O2-induced OS cell model used in this study. The authors stimulated cells with 1mM and 2mM H2O2 for 2,5h to induce OS. However, as shown in Fig2 and Fig3, viability of AAM-untreated cells is approximately 20-40%, which means that probably, 60-80% of cells are dead. It is quite difficult for cells recovering from that, especially because cell apoptosis in culture induces apoptosis per se. In the literature, we can found several studies using a maximum of 50-200 µM, depending on cell type. I recommend the authors to perform a dose-response curve for H2O2 for each cell line, evaluated ROS production, and then select a concentration of H2O2 sufficient to induce OS without killing more than 50% of the cells. AAM ability to protect or repair cells from OS should not be tested in conditions where only 20% of cells survive. Otherwise, the authors are not testing protection from OS, but from cell death (the authors said in line 532, that “dead cells no longer generate ROS”, which is obviously true, however it can induce others cells to die).
My other major concerns are as follows:
1. I do not understand why the cytotoxicity of AAM was measured in HeLa, HepG2 and A549 cancer cell lines (Figure1C,D,E), if the following experiments were only conducted in 3T3-L1 and HaCaT cells (Also correct the methods, lines 207-211, as it is stated the opposite).
2. The negative controls used in all experiments were “untreated cells” (line 200; 221), which are not correct. This is extremely important, because, as the authors recognized (line 424), methanol is cytotoxic and I am not sure if its removal from the extract was 100% efficient. Therefore, the negative control should be the vehicle, also subjected to the same methanol extraction method.
3. The authors performed pre and post-treatments with AAM extracts in H2O2-induced OS cellular models. Why they pre-treated 3T3-L1 and HaCat cell lines and post-treated 3T3-L1 only. Why they do not performed post-treatments in HaCat?
4. The authors used 2mM of H2O2 to test cytoprotective role of AMM extracts (Fig2,3), but then used 1mM and 2mM of H2O2 to evaluate ROS production (Fig4). Why introducing the concentration of 1mM?
5. Material and Methods needs to be revised: What are the time periods of AAM pre- and post-incubations? Did the authors washed the cells between incubations? How MTT and NRU were performed?
6. The authors should made clear the goal of the experiment of UV exposure. Is it to measure the ability of AAM to protect cells from UV-induced OS or to promote phototoxicity (Fig4)?
7. Maybe some new experiments regarding the activity or expression of antioxidant enzymes should be included. Otherwise, the experimental design of this article is a copy of another published by the authors regarding the Cytoprotective Properties of polypodium vulgare L. (doi.org/10.3389/fphar.2021.727528)
Minor points:
8. (lines 17-20) The abstract references to “in vitro activities” should be replaced for something more specific.
9. (line 62) “as a therapeutic tool as a” should be corrected.
10. (line 70) “a fact that increases their correct identification”, please revise, do not seems ok.
11. (lines 167-171) Spanish language is not acceptable.
12. (line 173) “wine” ?
13. (lines 356-358) Is it statistically different? If so, include asterisks in the graphs.
14. (lines 435-436) “The penetration of the extract into the mitochondria determines that it also penetrates the lysosomes that are in the cytosol…” Please explain.
15. (line 445) “a basic cell line” Please change to “a widely used cell line”
Author Response
Dear reviewer,
Thanks for your time and comments. Please find below a point-by-point our answers to your comments. In the revised manuscript, additions and corrections are highlighted in blue colour.
- Perhaps the major criticism I have relates to the H2O2-induced OS cell model used in this study. The authors stimulated cells with 1mM and 2mM H2O2 for 2,5h to induce OS. However, as shown in Fig2 and Fig3, viability of AAM-untreated cells is approximately 20-40%, which means that probably, 60-80% of cells are dead. It is quite difficult for cells recovering from that, especially because cell apoptosis in culture induces apoptosis per se. In the literature, we can found several studies using a maximum of 50-200 µM, depending on cell type. I recommend the authors to perform a dose-response curve for H2O2 for each cell line, evaluated ROS production, and then select a concentration of H2O2 sufficient to induce OS without killing more than 50% of the cells. AAM ability to protect or repair cells from OS should not be tested in conditions where only 20% of cells survive. Otherwise, the authors are not testing protection from OS, but from cell death (the authors said in line 532, that “dead cells no longer generate ROS”, which is obviously true, however it can induce others cells to die).
Answer: We agree with the reviewer the concentration of H2O2 must be adjusted to the objective of the study and cell type, but also depends on cell density and time incubation. Previous studies from our group demonstrated the protective effect of wine extracts and epicatechin conjugates in 3T3 and HaCaT, establishing the conditions of peroxide insult (Ugartondo et al., J Agric Food Chem 2009, 57(10):4459-65. doi: 10.1021/jf900240a; Mitjans et al., J Agric Food Chem 2011, 59(5):2113-9. doi: 10.1021/jf1025532). According to these studies, 3T3 and HaCaT showed a decrease in cell viability at 2 mM against H2O2 that allow to assay potential protection of AAM extract from cell death. Nevertheless, we have also assayed 1 mM H2O2 and we included in the supplementary material the results obtained. It can be observed that cytotoxicity induced by 1 mM H2O2 is less than 35% in both cell lines, but not protective effects were detected for AMM. In these experiments, cells were pre-treated with fern extract (24h) and cell viability was determined just after incubation with H2O2 (2,5 h). In these conditions, ROS generation can be assessed appropriately.
Viability of H2O2 treated cells presented in Figure 3 of the manuscript cannot be related to ROS experiments as cell viability was determined 24 hours after peroxide insult. Detection of ROS was performed during induction of oxidative stress, specifically at 1 hour, 2 and 3 hours.
My other major concerns are as follows:
- I do not understand why the cytotoxicity of AAM was measured in HeLa, HepG2 and A549 cancer cell lines (Figure1C,D,E), if the following experiments were only conducted in 3T3-L1 and HaCaT cells (Also correct the methods, lines 207-211, as it is stated the opposite).
Answer: We performed a comparative study among different non-tumoral and tumoral cell lines as an approach to understand the cytotoxicity behaviour of AAM together with the potential uses and applications. The reference studies to choose the cell lines were Alonso-Lebrero, et al., (J Photochem Photobiol B-Biology 2003, 70, 31-37, doi:10.1016/s1011-1344(03)00051-4), for the potential cutaneous applications, and Petkov et al., (Plants-Basel 2021, 10, 14, doi:10.3390/plants10061053), for the antiproliferative activity. However, our results did not support the last hypothesis. For this reason, posterior studies were only performed with 3T3 and HaCaT for the possible uses in cosmetic and/or pharmaceutical formulations. To clarify, the following sentence was added to section 3.2: For this reason, the following studies were only performed with non-tumoral cell lines.
Line 207 was rewritten according to our results.
- The negative controls used in all experiments were “untreated cells” (line 200; 221), which are not correct. This is extremely important, because, as the authors recognized (line 424), methanol is cytotoxic and I am not sure if its removal from the extract was 100% efficient. Therefore, the negative control should be the vehicle, also subjected to the same methanol extraction method.
Answer: The reviewer exposes that some amount of methanol remains in the extract inducing cell cytotoxicity and for this reason the negative control should be cells treated with the vehicle also subjected to the same methanol extraction method. Methanol is one of the preferred solvents because its volatility and high extraction efficiency which allows to completely remove it by evaporation. Then biological activities can be assessed by dissolving the dried methanolic extract in aqueous solutions. In our case, the extracts were subjected to rotatory evaporator to eliminate the methanol until dryness, so no methanol should remain in the extract. If traces of the solvent stand, it is reported that methanol at concentrations less than 2.5% were well-tolerated by cells (Nguyen et al., Biomed Res Ther 2020, 7(7):3855-3859. doi: 10.15419/bmrat.v7i7.614). Thus, in our cellular experiments, the different concentrations of AAM were freshly prepared in the hood dissolving the dry extract in culture media (5% FBS-DMEM). Therefore, our negative controls are the cells maintained with 5% FBS-DMEM or untreated cells. Culture media could not be subjected at the same process as the methanol extraction because we cannot guarantee the preservation of its characteristics and properties, which in turn can result in such a cell death that would be very difficult to evaluate cell cytotoxicity related to extract treatment.
- The authors performed pre and post-treatments with AAM extracts in H2O2-induced OS cellular models. Why they pre-treated 3T3-L1 and HaCat cell lines and post-treated 3T3-L1 only. Why they do not performed post-treatments in HaCat?
Answer: As observed in Figure 2 of the manuscript, we have only detected some cytoprotective activity for the AAM extract (pre-treatment) in the case of 3T3. Therefore, as Siddiqui et al. 2011 (doi: 10.3390/antiox9080704) reported, post-treatment presents good results only in the case that protective effects have been observed in the pre-treatment, since repairing mechanisms require more cellular energy than the protective ones. Consequently, we have only evaluated 3T3 in the cellular repair experiments.
- The authors used 2mM of H2O2 to test cytoprotective role of AMM extracts (Fig2,3), but then used 1mM and 2mM of H2O2 to evaluate ROS production (Fig4). Why introducing the concentration of 1mM?
Answer: Cytoprotective activity was also performed with cells treated with 1mM H2O2, as stated previously. In this sense, we enclosed in the supplementary material a Figure with the results obtained. As shown, cells treated with 1mM H2O2 for 2,5 hours show a viability of 71.0 ± 7.2 in the case of 3T3 and 66.3 ± 3.2 in the case of HaCaT. This high viability can account for the inexistence of a significant protective effect of the extract.
- Material and Methods needs to be revised: What are the time periods of AAM pre- and post-incubations? Did the authors washed the cells between incubations? How MTT and NRU were performed?
Answer: Thanks for the suggestion. We have included information about time, washing steps and performance of the MTT and NRU assays.
- The authors should made clear the goal of the experiment of UV exposure. Is it to measure the ability of AAM to protect cells from UV-induced OS or to promote phototoxicity (Fig4)?
Answer: OECD TG 432 (2019) describes the 3T3 neutral red uptake phototoxicity test, an in vitro assay widely used in pharmaceutic industry to identify potential phototoxicity induced by soluble compounds. Moreover, the induction of phototoxic reactions by plant extracts has been documented (Bark et al., J Ethnopharmacol 2010, 127: 11–18. doi: 10.1016/j.jep.2009.09.058 ; Fu et al., J Environ Sci Health, Part C 2013, 31:213–255. doi: 10.1080/10590501.2013.824206.) for this reason before to assay potential photoprotective activity of AAM we want to discard any adverse reaction due to concomitance of UVA light and the extract. In this sense, the following sentence was added to 2.7. section:
In this assay cytotoxicity of the extract was compared in the presence and in the absence of exposure to a non-cytotoxic dose of ultraviolet A light and, therefore, we can exclude the presence of phototoxic reactions.
In addition, in section 3.2.4., some other clarifications are included.
- Maybe some new experiments regarding the activity or expression of antioxidant enzymes should be included. Otherwise, the experimental design of this article is a copy of another published by the authors regarding the Cytoprotective Properties of polypodium vulgare L. (doi.org/10.3389/fphar.2021.727528)
Answer: This study is part of a general one. The main goal is to characterize the most representative fern of a region and to relate to their traditional and/or potential uses. We agree that those new determinations and evaluations could contribute to a better understanding of the antioxidant activity of the fern but here we present a preliminary study.
Minor points:
- (lines 17-20) The abstract references to “in vitro activities” should be replaced for something more specific.
Answer: Thanks for the suggestion, we have included the specific activities assessed.
- (line 62) “as a therapeutic tool as a” should be corrected.
Answer: We have rewritten the text.
- (line 70) “a fact that increases their correct identification”, please revise, do not seems ok.
Answer: The sentence was revised and removed accordingly. Moreover, introduction was shortened according to reviewer 2 suggestion.
- (lines 167-171) Spanish language is not acceptable.
Answer: We apologise for the mistake; We have rewritten the text in the correct language.
- (line 173) “wine”?
Answer: We apologise for the mistake, and we have replaced “wine” for “sample”.
- (lines 356-358) Is it statistically different? If so, include asterisks in the graphs.
Answers: Corrections according to the comments are included in the text.
- (lines 435-436) “The penetration of the extract into the mitochondria determines that it also penetrates the lysosomes that are in the cytosol…” Please explain.
Answer: The sentence was rewritten and corrected.
- (line 445) “a basic cell line” Please change to “a widely used cell line”
Answers: The suggested change was included.
We want to thank you the reviewer for its constructive criticisms and suggestions.
Reviewer 3 Report
The manuscript made by Farras et al. is focused on the study of the potential of polyphenolic extract of Asplenium adiantum nigrum as a cytoprotective agent with the effect against oxidative stress. The manuscript is written well.
I have just a few recommendations:
Introduction:
In the first three paragraphs, you mentioned general information about phnolic compounds as important agents wtih effect against oxidative stress. You have used some information about traditional medicine and ethnomedicine. I think, that you can cite more recent studies especially in the third paragraph. For example Fatima et al. Phytochemicals from Indian Ethnomedicines: Promising Prospects for the Management of Oxidative Stress and Cancer. Antioxidants. 2021.
Materials and Methods:
subchapter 2.2: You have mentioned that you dried specimen of the plants. Could you add information the conditions of drying?
Conclusion:
Last sentence: "...with potential pharmaceutical applications." Could you suggest which pharmaceuticals applications you mean?
I recommend to accept the manuscript after suggested minor revisions.
Author Response
Dear reviewer,
Thanks for your time and comments. Please find below a point-by-point our answers to your comments. In the revised manuscript, additions and corrections are highlighted in blue colour.
- Introduction:
- In the first three paragraphs, you mentioned general information about phnolic compounds as important agents wtih effect against oxidative stress. You have used some information about traditional medicine and ethnomedicine. I think, that you can cite more recent studies especially in the third paragraph. For example Fatima et al. Phytochemicals from Indian Ethnomedicines: Promising Prospects for the Management of Oxidative Stress and Cancer. Antioxidants. 2021.
Answer: We have included the reference of Fatima et al., (Antioxidants 2021, 10(10), 1606; doi: 10.3390/antiox10101606) in the introduction and discussion.
- Materials and Methods:
- subchapter 2.2: You have mentioned that you dried specimen of the plants. Could you add information the conditions of drying?
Answer: The following sentence is added to subchapter 2.2: The fronds were dried at room temperature under laboratory filter paper for 4-5 consecutive days and, finally, a …
- Conclusion:
- Last sentence: "...with potential pharmaceutical applications." Could you suggest which pharmaceuticals applications you mean?
Answer: According to the request of reviewer, we have included some potential applications of the fern extract.
- I recommend to accept the manuscript after suggested minor revisions.
We want to thank you the reviewer for its constructive criticisms and suggestions. We are happy that the reviewer finds our study interesting and suitable for the Journal.
Round 2
Reviewer 2 Report
I have read with care the revised version and rebuttal letter by Farràs et al.. Thank you for the answers, but unfortunately the work still needs further analysis and adequacy of the results presented.
Although some of the criticisms have been addressed by the authors, I am still not convinced about the originality of this study and the reliability of the results. I think that the objective of the study is not clear, and the methodology is also confusing. It needs some experiments to support discussion and conclusion of the results.
1. The justification for not using a vehicle as a negative control is not valid. I was not expecting to use culture media subjected to methanol extraction as a vehicle. It is supposed to use H2O, for example, in a similar volume of the powdered fronds, subjected to methanol extraction and all the next steps, following the same process as for the AAM extract. Then, this vehicle (H2O) would be diluted in 5% FBS-DMEM to stimulate cells, similarly to AAM extracts.
2. The reason why the post-treatments with AAM were conducted only in 3T3-L1, even though the pre-treatments were conducted in both 3T3-L1 and HaCat cell lines, should be clear in the text.
3. I have doubts on the adequacy of phototoxicity test in this study. With this test, the authors want to discard “any adverse reaction due to concomitance of UVA light and the extract”. However, UV exposure can induce oxidative stress (OS) by itself; and AAM extract can potentially protect cells from OS; so the result of this test would be a balance between its OS protective role and its phototoxicity activity…too confusing.
4. Material and Methods needs further descriptions related to the time periods of H2O2 and AAM treatments before MTT and ROS assays. The authors said that “Figure 3 of the manuscript cannot be related to ROS experiments as cell viability was determined 24 hours after peroxide insult. Detection of ROS was performed during induction of oxidative stress, specifically at 1 hour, 2 and 3 hours.” However, in the article we can read “cells treated with 2 mM H2O2 during 2.5 h” for cell viability and “Intracellular ROS induced by 1 and 2 mM H2O2 for 2 h treatment with AAM” for ROS detection.
5. I understand that this study is a preliminary one, but, still, it needs to be clear in the objectives and present the proper methodology.
6. How the authors explain the results? The protective role was observed only in 3T3-L1 cells treated with 2mM H2O2 and 1mg/ml AAM, only when approximately 20-30% of cells remain alive.
-Why this effect was not observed when viability is around 70%, where H2O2 insult (1mM) was also significative?
-Why this effect was confined to 3T3-L1 at concentration of 1mM of AAM, and why ROS was only detected in HaCat cells?
-Authors cannot conclude that AAM mitigates oxidative stress based on an isolated result. Additional experiments are needed to validate this hypothesis.
Author Response
Dear reviewer,
We appreciate the time and effort that you have dedicated to providing your valuable feedback on our manuscript. We are grateful for the insightful comments on our paper. We have been able to incorporate changes to reflect most of the suggestions provided by the reviewer. We have highlighted the changes within the manuscript in red.
Here is a point-by-point response to the reviewers’ comments and concerns.
- The justification for not using a vehicle as a negative control is not valid. I was not expecting to use culture media subjected to methanol extraction as a vehicle. It is supposed to use H2O, for example, in a similar volume of the powdered fronds, subjected to methanol extraction and all the next steps, following the same process as for the AAM extract. Then, this vehicle (H2O) would be diluted in 5% FBS-DMEM to stimulate cells, similarly to AAM extracts.
Answer: We apologize about our misunderstanding. We understand “vehicle” as the solvent used to dissolve the extract before its application on the cells. Now, as we understand, the reviewer asks why cells treated with “vehicle used for the extraction” diluted in the culture medium is not the control used for comparative purposes instead of the mandatory untreated cells (actual control of normal cell proliferation and growth). We have not used methanol (or H2O subjected to extraction process) as control (blank?) in our experiments because the extract does not contain this solvent. Methanol is widely used in the field of medicinal plants to extract bioactive compounds and polyphenols because it is relatively cheap and easy to remove with a rotatory evaporator (Plants (Basel), 2021. 10(10):2091. doi: 10.3390/plants10102091). After performing the maceration of the plant with the solvent, methanol was removed by subjecting the extract at 37 ºC under vacuum for 8 h in a rotatory evaporator. This is a common procedure working with plant extracts. Additionally, as it can be observed in Figure 1, no cytotoxic effects were detected at the maximum tested concentration (1 mg/mL) in the different cell lines, which means that the extract is methanol free; in the case that there were traces of methanol they do not affect to cell viability as shown in Figure 1. The maximum tested concentration (1 mg/mL) is a high concentration of extract in the wells so we can conclude that this extract is quite "safe" under these conditions.
- The reason why the post-treatments with AAM were conducted only in 3T3-L1, even though the pre-treatments were conducted in both 3T3-L1 and HaCat cell lines, should be clear in the text.
Answer: As requested, the reasons why cellular repair was only performed in NIH 3T3 cells was included in section 3.2.2. According to Siddiqui et al. 2011, cellular repair activity (i. e. post-treatment by the extract) presents good results only in the case that protective effects have been observed in the pre-treatment, since repairing mechanisms require more cellular energy than the protective ones. Consequently, cellular repair activity was only assessed in 3T3 cells.
- I have doubts on the adequacy of phototoxicity test in this study. With this test, the authors want to discard “any adverse reaction due to concomitance of UVA light and the extract”. However, UV exposure can induce oxidative stress (OS) by itself; and AAM extract can potentially protect cells from OS; so the result of this test would be a balance between its OS protective role and its phototoxicity activity…too confusing.
Answer: Phototoxicity is an acute light-induced response, which occurs when photoreactive chemicals are activated by solar light and transformed into products cytotoxic against the skin cells (Kim et al., 2015. Toxicol Res. 31(2):97-104. doi: 10.5487/TR.2015.31.2.097). Polyphenols are widely used in cosmetic formulations for its beneficial properties, but they are characterized by the presence of a benzene ring (Leopoldini et al., 2011. Food Chemistry, 125, 288-306, doi:10.1016/j.foodchem.2010.08.012), a molecular structure responsible in the induction of phototoxic reactions. In this sense, the use of 3T3 NRU assay (OECD TG 432, 2019) has been used in different studies to evaluate the potential phototoxic activity of different antioxidant extracts (Amaral et al., 2014. BMC Complement Altern Med, 18;14:450. doi: 10.1186/1472-6882-14-450; Napoli et al., 2018. Phytochemistry, 152: 162-173. doi: 10.1016/j.phytochem.2018.05.003; Fernandes et al., 2019 BMC Pharmacol Toxicol, 20(1):77. doi: 10.1186/s40360-019-0353-3; Baccarin et al., 2015. J Photochem Photobiol B, 153: 127-36. doi: 10.1016/j.jphotobiol.2015.09.005) as we reported here for AAM. Moreover, recently, the phototoxic potential of quercetin is demonstrated in 3T3 and HaCaT cells (Svobodová et al., 2017. Photochem Photobiol, 93(5):1240-1247. doi: 10.1111/php.12755), while the non-phototoxic capacity of caffeic acid or rutin is also described (Aguiar et al., 2021. Molecules, 27(1): 189. doi: 10.3390/molecules27010189). Therefore, potential antioxidant activity is noninterfering in those cellular in vitro phototoxic assay and identification of both phototoxic and non-phototoxic agents is possible. Here we have used an adaptation of the 3T3 NRU phototoxicity assay and no significant decrease of cell viability was recorded in the case of exposed cells to AAM, and UVA light compared to cells only exposed to AAM.
- Material and Methods needs further descriptions related to the time periods of H2O2 and AAM treatments before MTT and ROS assays. The authors said that “Figure 3 of the manuscript cannot be related to ROS experiments as cell viability was determined 24 hours after peroxide insult. Detection of ROS was performed during induction of oxidative stress, specifically at 1 hour, 2 and 3 hours.” However, in the article we can read “cells treated with 2 mM H2O2 during 2.5 h” for cell viability and “Intracellular ROS induced by 1 and 2 mM H2O2 for 2 h treatment with AAM” for ROS detection.
Answer: we have rewritten the explanations of Figure 2, 3 and 5 of the revised manuscript as follows:
Figure 2: Cytoprotective activity of AAM in 3T3 (A) and HaCaT (B). Cells were incubated 24 h in the absence (white column) or in the presence of AAM (gray columns) and subsequently treated with 2 mM H2O2 for 2.5 h and finally cell viability was determined by MTT assay and expressed as percentage respect to untreated cells. H2O2 cell viability in absence of the extract was used as positive control.
Figure 3: Cellular repair activity of AAM in 3T3 cells. Cells previously treated with 2 mM H2O2 for 2.5 h were incubated in the absence (white column) or in the presence of AAM for 24 h more (gray columns) and finally cell viability was determined by MTT assay and expressed as percentage respect to untreated cells. H2O2 cell viability in absence of the extract was used as positive control.
Figure 5: Intracellular ROS induced by 1 and 2 mM H2O2 for 2 h treatment after 24 h incubation with AAM in 3T3 (A) and HaCaT (B) cells
- I understand that this study is a preliminary one, but, still, it needs to be clear in the objectives and present the proper methodology.
Answer: we have rewritten the objectives.
The main goal of our work is to characterize one of the most representative fern of Prades mountains (Tarragona, Spain) and to relate to their traditional and/or potential uses. As a preliminary study, we present here the characterization of a polyphenolic extract obtained from the fronds of Asplenium adiantum-nigrum L. analyzing 38 phytochemicals. Moreover, the cytotoxic profile in various cell lines, their potential cytoprotective and cellular repair activity, phototoxic activity and protection against ROS production was assessed in 3T3 and HaCaT.
- How the authors explain the results? The protective role was observed only in 3T3-L1 cells treated with 2mM H2O2 and 1mg/ml AAM, only when approximately 20-30% of cells remain alive.
-Why this effect was not observed when viability is around 70%, where H2O2 insult (1mM) was also significative?
-Why this effect was confined to 3T3-L1 at concentration of 1mM of AAM, and why ROS was only detected in HaCat cells?
-Authors cannot conclude that AAM mitigates oxidative stress based on an isolated result. Additional experiments are needed to validate this hypothesis.
Answer: We really appreciate the comments and suggestions of the reviewer, but this manuscript is part of a PhD thesis that has been already concluded and it not possible to continue making experiments at this point due to lack of financial support. However, we will consider more experiments in the future if we continue doing research on this extract to confirm potential pharmaceutical applications. The interest of Asplenium adiantum-nigrum is based on the content of polyphenols and absence of toxicity in different cell lines which makes this species a valuable source of bioactive compounds of phenolic origin. To the best of our knowledge, this is the first work in which 38 phenolic compounds of different classes are analyzed in this species, contributing to the field. In fact, this manuscript has been submitted to a Special Issue entitled "Biological activities of medicinal and aromatic plants" devoted to exploring extracts with bioactive compounds.

Round 3
Reviewer 2 Report
Although I think the experimental data could have been improved, I understand the limitations of this study. All my criticisms have been answered by the authors and the manuscript has been amended accordingly. I totally agree with the change of the title.